# In-Situ Approaches for the Preparation of Polythiophene-Derivative Cellulose Composites with High Flexibility and Conductivity

Francisco González, Pilar Tiemblo and Mario Hoyos *

Instituto de Ciencia y Tecnología de Polímeros, ICTP-CSIC, Juan de la Cierva, 3, 28006 Madrid, Spain
* Correspondence: hoyos@ictp.csic.es

**Abstract:** Composite materials of conjugated polymers/cellulose were fabricated by incorporating different polythiophene-derivative polymers: Poly(3,4-ethylenedioxythiophene) (PEDOT) and an alkylated derivative of poly(3,4-propylenedioxythiophene) (PProDOT). These conjugated polythiophenes were deposited by casting or spray coating methodologies onto three different cellulose substrates: Conventional filters papers as cellulose acetate, cellulose grade 40 Whatman® and cellulose membranes prepared from cellulose microfibers. The preparation of composite materials was carried out by two methodologies: (i) by employing in-situ polymerization of 3,4-ethylenedioxithiophene (EDOT) or (ii) by depositing solutions of poly(3,4-ethylenedioxythiophene):polystyrene sulfonate (PEDOT:PSS) or lab-synthetized PProDOT. Composite materials were studied in terms of electrical conductivity and surface morphology assessed by impedance spectroscopy, surface conductivity, SEM, and 3D optical profilometry. In-situ composite materials prepared by spray coating using iron trifluoromethane sulfonate as oxidizing agent can be handled and folded as the original cellulose membranes displaying a surface conductivity around $1 \text{ S·cm}^{-1}$. This versatile procedure to prepare conductive composite materials has the potential to be implemented in flexible electrodes for energy storage applications.

**Keywords:** conjugated polymers; in-situ composite materials; conductivity; 3D optical profilometry; flexible electrodes; energy storage

## 1. Introduction

Intrinsically conducting polymers were first time synthetized by Heeger, MacDiarmid and Shirakawa in the 1970's decade of the last century [1]. Their electrical properties are due to their chemical structure presenting a conjugated bond system along the polymeric chain, reason why they are denoted as conjugated polymers (CPs) where its electrical properties can be increased several magnitude orders by a doping process. Among them, polypyrrol (PPy); polyaniline (PANI); and thiophene-derived polymers such as poly(3-hexylthiophene) (P3HT), poly(3,4-ethylenedioxythiophene) (PEDOT), and poly(3,4-propylenedioxythiophene) (PProDOT) are the most employed due to their electrical properties and stability. Materials based on CPs are of particular interest due to the wide range of diverse applications that have been developed in organic electronics as antistatic agents, OLEDs, transistors, organic solar cells, electrochromic devices, and energy storage applications as supercapacitors [2,3].

Practical advantages of PEDOT include possibilities to achieve high charge storage capacities, lightweight, good processability, as well as high conductivities [4,5]. Materials based on PEDOT are of particular interest due to their high electrical conductivity, mechanical flexibility, large electroactive potential window, and reasonable theoretical capacitance [3], as well as good stability comparing with other CPs [6–8].

The drawback of PEDOT-based supercapacitor electrodes lies mainly in a classical deposition methodology, such as electrochemical or on vapor phase, since long times and high temperatures are needed, resulting in relatively expensive manufacturing processes. These methods also typically yield electrodes with PEDOT layers with a thickness of the order of tens microns to several hundred nanometers, which exhibit low capacitances to be appropriate for up-scaled production in industry [9]. This behavior typically corresponds to a thin electroactive layer of PEDOT on the surface with low surface area or thick layers, which inhibit fast electrochemical access. Therefore, to obtain a supercapacitor with high charge storage capacity, it is hence important to use electrode materials with thin PEDOT layers and large surface areas.

The basic structural motif of PEDOT, the alternating arrangement of single and double bonds along the backbone, gives rise to the band structure but also restricts conformational freedom of unsubstituted conjugated and other rod-like polymers [10]. These materials are thus typically insoluble and rigid. The rigidity of all-$sp^2$-hybridized materials originally made CPs attractive for their mechanical strength [11,12]. For flexible electronic devices, however, tensile strength is less important than is elasticity and toughness [13], which contributes to the robustness of thin devices. Hence, novel scalable approaches to fabricate flexible and conducting PEDOT-derived materials with high capacitances are therefore required.

An encouraging approach for transforming PEDOT into a flexible material without altering its inherent conductive properties is making paper-like materials electrically conductive for later use in energy storage devices [14]. The development of light, flexible and low-cost supercapacitors made from cellulose is a promising approach as it can take advantage of the porous paper structure as substrate, due to its large surface roughness and specific area. To obtain these cellulose-composite materials, different strategies have been employed, for example, coating the paper with a conductive matrix like carbon nanotubes, graphene, conducting polymers, graphite, etc. These conductive materials are added using different techniques such as 'soak and dry', 'paper making', vacuum filtration, simple drawing coatings methodologies [14,15]. In terms of scalability, 'soak and dry' coating techniques as spray coating and drop casting are promising technologies, whose objective is to obtain a stable coating by the permeation of conductive particles through cellulose fibers. CPs can be dissolved in eco-friendly solvents like water, commonly used in the case of poly(3,4-ethylenedioxythiophene)-poly(styrenesulfonate) (PEDOT:PSS). Another interesting approach is to perform the in-situ polymerization, which enables to polymerize onto the cellulose fibers [16]. The source of cellulose is important, as it could have different interactions with solvent and CPs, and in consequence different properties, such as its flexibility and conductivity [17]. Despite the insulating behavior of cellulosic substrates, it has been shown that conjugated polymer-cellulose composite electrode materials feature high capacitances and conductivity and low sheet resistances [18]. Recently, Nyholm and co-workers reported the use of cellulose nanofibers as building blocks on the fabrication of nanostructured PEDOT flexible paper [16]. The research group has described a versatile solution for the development of highly conductive and flexible nanocellulose fibers coated with nanostructured PEDOT within 30 min leading to a material with a large surface area, low sheet resistance, and high active mass loading (7.3 mg·cm$^{-2}$) and high specific electrode capacitances (0.92 F·cm$^{-2}$).

The relationship between conductive properties and surface topography of cellulose composite materials is a fundamental issue as surface properties may have a fundamental impact on the performance of paper-based supercapacitors [14]. However, to characterize this relationship is not always an easy task as difficulties are reported with the measuring of similar materials with AFM due to the non-uniform geometrical structure and size of cellulose fibers [19].

In this work, we describe a straightforward, fast and versatile solution-based approach for the fabrication of robust, conductive and highly flexible PEDOT paper. Within our research group, we have already developed a methodology for the preparation of highly luminescent and superhydrophobic polyfluorene/silica/cellulose hybrid materials. These multifunctional, versatile, and easily processed and scalable hybrid materials are suitable for large surfaces and industrial applications [20,21].

To our knowledge, surface properties of CP-cellulose materials have not been studied with 3D optical profilometry. Among other advantages, 3D optical profilometry allows to obtain fast 3D real-color imaging with a vertical and lateral resolution more adequate to characterize cellulose microfibers composite materials than AFM. We present the process for obtaining conductive paper by casting or spray coating different thiophene-derived polymers as PEDOT and a soluble in organic solvents alkylated PProDOT. Moreover, we study the effect of different substrates (filter conventional paper, cellulose acetate and isolated cellulose microfibers) on the free-standing PEDOT structure employing different processing methodologies that were used for polythiophenes-derivative polymers deposition of conducting papers. Towards a better understanding of properties and interactions between cellulose and CPs, thermal stability, infrared spectroscopy, scanning electron microscopy, electrical conductivity, and surface 3D optical profilometry have been carried out. Composite materials were evaluated in terms of electrical conductivity and surface roughness, obtaining a dependence of the electrical properties with the surface morphology and processing conditions (solvent and nature of the substrate). Regardless the surface roughness of the cellulosic substrate, it is possible to produce flexible composite materials with good conductive properties through in-situ approach in short times.

## 2. Materials and Methods

### 2.1. Materials

EDOT, PEDOT:PSS (3 wt.% in $H_2O$, conductive grade), iron (III) tosylate hexahydrate (Fe(Tos)$_3$), iron(III) trifluoromethane sulfonate (Fe(Trif)$_3$), and chemicals used for synthesis: Pluronic® F-127, imidazole, dimethoxythiophene, p-toluene sulfonic acid, diethyl malonate, hexyl bromide, sodium hydride, lithium-aluminium hydride, and anhydrous toluene and tetrahydrofuran solvents were purchased from Sigma-Aldrich. From Alfa-Aesar, 2,2-dibutylpropane-1,3-diol was purchased. Ethanol, methanol, acetonitrile, n-metilpirrolidone (NMP) and dichloromethane (DCM) used as solvents were of reagent grade. Cellulose acetate esters filter membranes (A) were obtained from HAWP Millipore® (0.45 μm); filter paper of grade 40 (8 μm) was provided from Whatman® (W); cellulose microfibers from kraft pulp were kindly provided from Stora Enso Innovation Centre, Sweden.

### 2.2. Preparation of Cellulose-Polythiophene Materials

#### 2.2.1. Cellulosic Substrates

A and W filter papers were cut into discs of 20 mm of diameter. On their turn, cellulose fibers were dispersed in water by magnetic stirring at a concentration of 0.5 wt.% for 24 h, filtered under vacuum, and then dried at room temperature overnight. Afterwards, filtrated membranes were pressed at room temperature using a hydraulic press Specac® at a pressure of 2 tons to obtain cellulose microfibers membrane (C).

#### 2.2.2. Synthesis of Monomers and Polymerization

PEDOT:Tosylate and PEDOT:Triflate. Synthesis of PEDOT using Fe(Tos)$_3$ was synthesized adapting a reported procedure [22]. A mixture of Fe(Tos)$_3$ (20.47 g) and imidazole (2.14 g) in 55 mL of methanol was added drop by drop to a solution of 2.2 mL of EDOT dissolved in 55 mL of methanol. The polymerization was performed at 50 °C after 25 min. The product was washed with methanol and then dried under vacuum overnight (99%, 1.9 g). Synthesis of PEDOT:Triflate was carried out following a reported procedure [23] by preparing a solution of 20 wt.% of the surfactant Pluronic® F-127 in ethanol (solution 1) and a solution of 20 wt.% of the surfactant in NMP (solution 2). To a mixture of 5.7 g of solution 1 and 0.6 g of solution 2 (0.7 wt.% of surfactant), 1 g of Fe(Trif)$_3$ was added, and the resulting solution was sonicated during 3 h. Throughout this time, the solution changed from orange to dark red. Subsequently, 0.15 mL of EDOT (0.2 g) was added dropwise into the solution in the ultrasonic agitator with a molar relationship monomer:oxidant of 1:1.27 to obtain PEDOT:Triflate.

Sonication continued for one more minute, and the dark solution was put in a crystallizer at 70 °C. The product was thoroughly washed with ethanol to obtain the final black product (99%, 0.584 g).

PProDOT. Synthesis of 2,2-dibutyl-3,4-propylenedioxithiophene (DBProDOT) was performed as follows based on reported literature [24,25]. A total of 0.25 g of synthetized monomer was polymerized using a 40 wt.% solution of Fe(Tos)$_3$ (0.6 mol·L$^{-1}$) in ethanol with a molar relationship monomer-oxidant of 1:2 to obtain PDBProDOT in 10 min. The polymer was precipitated into methanol and extracted into a Soxhlet thimble for 24 h with methanol and then with hexanes. A subsequent Soxhlet extraction with DCM was used to dissolve the polymer from the thimble and the solvent was removed under vacuum. Poly(3,3-Dibutyl-3,4-dihydro-2H-thieno [3,4-b][1,4]dioxepine) [PDBProDOT]. Yield: 0.29 g of deep purple solid (99%). $^1$H-NMR (400 MHz, CDCl$_3$): δ 7.77 (s, Hterminal), 3.85 (s, 4H), 1.64 − 1.0 (m, 12H), 0.99 − 0.69 (m, 6H). GPC analysis: Mn: 4800 g·mol$^{-1}$; Mw: 15,360 g·mol$^{-1}$; PDI: 3.2.

### 2.2.3. Polythiophenes-Derivative/Cellulose Composite Materials: In-Situ Polymerization

To prepare polythiophene-derivative composite materials, 0.1 mL of EDOT (0.16 mol·L$^{-1}$) monomer was added dropwise on the cellulose substrate, for the subsequently addition of 2 mL of a solution of Fe(Tos)$_3$ or Fe(Trif)$_3$ (0.27 mol·L$^{-1}$) either by casting or by spray coating. Afterwards, samples were placed on a heating plate at 50 °C during 1 h for casting samples. Composite materials prepared by spray coating were dried at room temperature overnight. After that, samples were thoroughly washed with methanol to remove reaction sub-products, starting materials, and iron traces to obtain PEDOT cellulose membranes. Details of the samples prepared appear in Table 1.

**Table 1.** Polythiophene-derivative/cellulose materials prepared by in-situ polymerization.

| *In-Situ* Polymerization | | | |
|---|---|---|---|
| **Sample** | **Substrate** | **Proc. Technique** | **Oxidizing Agent** |
| PTsWCt | W | | |
| PTsACt | A | Casting (Ct) | |
| PTsCCt | C | | Fe(Tos)$_3$ |
| PTsWSp | W | | |
| PTsASp | A | Spray (Sp) | |
| PTsCSp | C | | |
| PTfWCt | W | | |
| PTfACt | A | Casting (Ct) | |
| PTfCCt | C | | Fe(Trif)$_3$ |
| PTfWSp | W | | |
| PTfASp | A | Spray (Sp) | |
| PTfCSp | C | | |

### 2.2.4. Polythiophenes-Derivative/Cellulose Composite Materials: Conjugated Polymers Solutions

For a better understanding of interactions between CPs and cellulose fibers, we also prepared composite materials processing CPs solutions, PEDOT:PSS and PDBProDOT, to perform a comparison on processability, roughness, and conductivity properties with in-situ prepared composite materials. A total of 2 mL of PEDOT:PSS aqueous solution was deposited on cellulose substrates either casting of spray coating processing methods. All samples were dried overnight at room temperature to evaporate solvent (Table 2). PDBProDOT cellulose composite materials were prepared through a similar procedure to PEDOT:PSS samples dissolving previously PDBProDOT in DCM by magnetic stirring during 1 h (Table 2). Additionally, Scheme 1 shows a detailed procedure for the preparation of all composite materials.

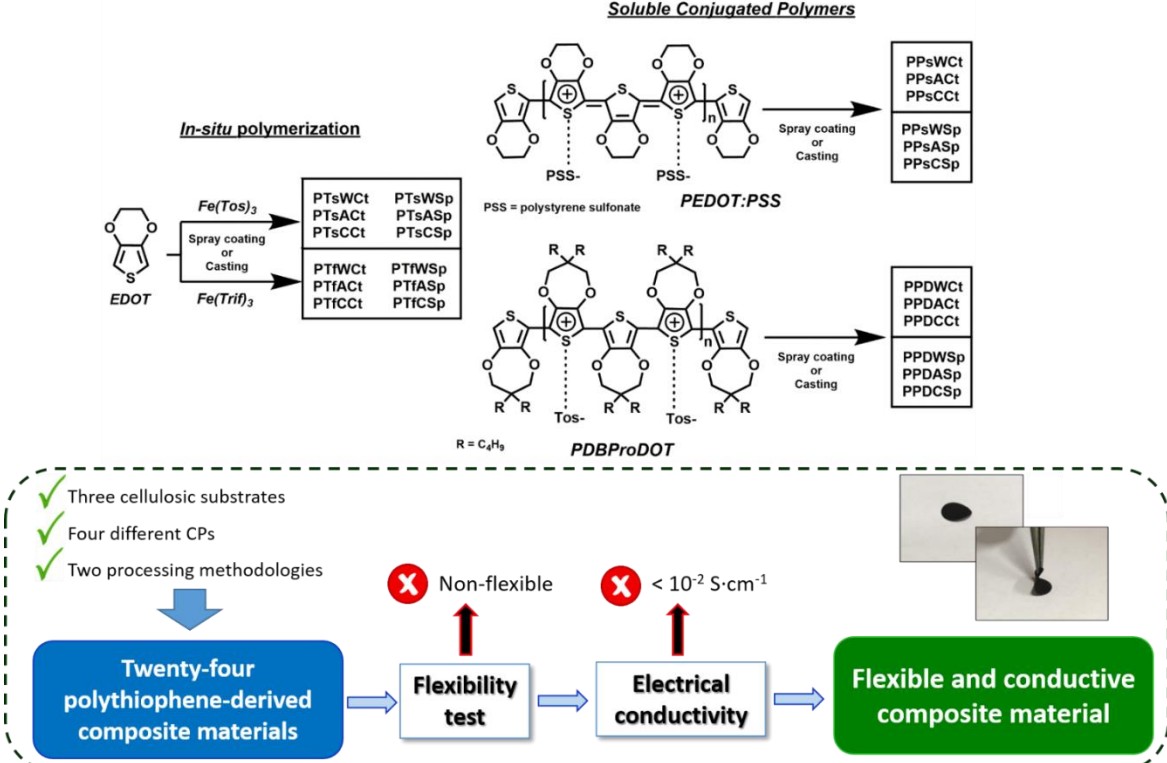

**Scheme 1.** Materials preparation by in-situ polymerization and soluble conjugated polymers. Followed methodology for the selection of flexible and conductive PEDOT-derived composite materials.

In this work, composite materials were named by each polymer name ('PTs', 'PTf', 'PPs' and 'PPD' correspond to PEDOT:Tosylate, PEDOT:Triflate, PEDOT:PSS and PDBProDOT, respectively) followed by the substrate type by their corresponding initials: W, A and C. The sample names are completed adding at the end the processing technique, casting (Ct) or spray coating (Sp).

**Table 2.** Polythiophene-derivative/cellulose materials prepared by solution.

| Conjugated Polymers Solution | | | |
|---|---|---|---|
| **Sample** | **Substrate** | **Proc. Technique** | **CPs *** |
| *PPsWCt* | W | | |
| *PPsACt* | A | Casting (Ct) | PEDOT:PSS |
| *PPsCCt* | C | | |
| *PPsWSp* | W | | |
| *PPsASp* | A | Spray (Sp) | PEDOT:PSS |
| *PPsCSp* | C | | |
| *PPDWCt* | W | | |
| *PPDACt* | A | Casting (Ct) | |
| *PPDCCt* | C | | PDBProDOT |
| *PPDWSp* | W | | |
| *PPDASp* | A | Spray (Sp) | |
| *PPDCSp* | C | | |

\* Polymers were all prepared into a 1.5 wt.% solution.

### 2.3. Characterization

*Nuclear Magnetic Resonance ($^1$H-NMR)* was performed in an Avance™ AV Ultrashield spectrometer from Bruker Coorporation (Billerica, MA, USA), with a frequency of 400 MHz. All samples were

dissolved in deuterated chloroform (CDCl$_3$, residual reference peak δ = 7.26 ppm). Polymer molecular weight was determined by *Gel Permeation Chromatography (GPC)* using a W1515 Isocratic HPLC Pump from Waters Corp. (Milford, MA, USA). Measurements were carried out in tetrahydrofuran (THF), 99.5+% solution stabilized with 250 ppm of BHT at 35 °C, and at a flow rate of 1 mL·min$^{-1}$ using Styragel Water columns (300 × 7.8 mm; 5 µm nominal pore size) and a W2414 Refractive Index detector from Waters. The system was calibrated with low polydispersity polystyrene standards in the range of 200 to 200 × 10$^3$ g·mol$^{-1}$. *ATR-FTIR* spectra were recorded using a Spectrum Two from Perkin Elmer, Inc. (Waltham, MA, USA) with 4 scans and a resolution of 4 cm$^{-1}$. *Thermogravimetric Analysis (TGA)* measurements were carried out in a TA Q500 from TA Instruments (New Castle, DE, USA) under air atmosphere at a heating rate of 10 °C·min$^{-1}$ up to 700 °C. *Scanning Electron Microscopy (SEM)* analysis was recorder using a Hitachi SU-8000 (Tokyo, Japan). For cross-sectional micrographs samples were fractured after immersion in liquid nitrogen and the sections were observed metalized.

*3D Optical profiles* measurements were obtained using a Zeta Instruments Zeta-20 3D profiler from KLA-Tencor (Milpitas, CA, USA), with a 50× optical objective and 13 nm of vertical resolution. Images were recorded using the real color mode. Areal surface roughness, defined as the arithmetical mean height of the surface (Sa), was obtained in at least three different regions of the samples observed using the Zeta 3D Software (v1.9.5.6.). The *electrical conductivity* was determined by two methods: *bulk conductivity (σ)* was measured by electrochemical impedance spectroscopy (EIS) using a Concept 40 broadband dielectric spectrometer from Novocontrol Technologies GmbH & Co. KG (Montabaur, Germany) in the frequency range between 0.1 and 10$^7$ Hz. Samples of 20 mm of diameter were placed between two gold-plated flat electrodes applying a voltage of 20 mV, and σ was determined in the non-frequency dependent behavior in graph. *Surface conductivity (σ$_s$)* was measured by the 4-point probe method, employing a current range of 1–10 mA taking into account the shape and thickness correction factors [26] with a DC low-current source (LCS-02) and a digital micro-voltmeter (DMV-001) from Scientific Equipment & Services, Roorkee, India. To determine σ and σ$_s$ of pure polymers, samples were previously compressed on pellets of 13 mm of diameter.

*Flexibility test*: The flexibility test consisted of bending the material in different parts more than twenty times. The materials that showed stiffness, brittleness, or had lost a loss of polymeric material during or after the test, were automatically discarded before performing the electrical measures. Therefore, electrical characterization was measured after the flexibility tests in all materials.

## 3. Results and Discussions

### 3.1. Polymerization

PEDOT:Tosylate and PEDOT:Triflate were successfully prepared within 25 min at 50 °C by redox polymerization of EDOT using a concentration of 0.27 mol·L$^{-1}$ of a solution of Fe(Tos)$_3$ or Fe(Trif)$_3$. The concentration ratio EDOT:oxidant resulted suitable for the polymerization of CPs in high yields (>99%). Polymerization time turned out to be shorter than 15 min; the choice of a low boiling point solvent as methanol for EDOT polymerization leads to a lower viscosity medium, faster reaction kinetics, and improved packaging of polymer chains [27,28]. A fine deep blue powder was obtained that was thoroughly washed with methanol and acetonitrile and filtered through nylon membranes (0.45 µm) in order to remove sub-products, starting materials, and iron traces.

### 3.2. Synthesis of PDBProDOT

Polymerization was highly dependent on the monomer concentration in the case of materials prepared by dispersion on cellulosic substrates. Polymers with moderate molecular weight were found in high yield when concentrations of 0.16 mol·L$^{-1}$ were used. Polymer yields above 95% (before purification) were obtained in all cases using a 0.65 mol·L$^{-1}$ concentration of Fe(Tos)$_3$ as oxidant. The use of this concentration of oxidant resulted in moderate molecular weight in very short reaction times obtaining unimodal molecular weight distributions after polymer purification. Polymer

purification was carried out by Soxhlet extraction during 24 h with sequential washing using methanol and hexanes. This procedure led to lower polydispersity and higher number average molecular weights (Mn) for the isolated polymers, which could be recovered in around 60% final yield. Figures S1 and S2 show the ${}^{1}$H-NMR spectrum and GPC analysis, respectively, of the successful synthesis and purification of PDBProDOT. The final polymers showed highly solubility in THF and DCM.

### 3.3. Preparation of Polythiophene-Derivative/Cellulose Composite Materials

As presented in Scheme 1, four different CPs, three cellulosic substrates, and two processing techniques were evaluated to obtain twenty-four composite materials. Figure 1 shows photographs of all the materials prepared in this work compared to the cellulosic substrates tested. When possible, photographs show the ability of the samples to be bended without damage just as original cellulosic substrates. From materials prepared by in-situ polymerization, it can be noted than those prepared using A substrates (PTsACt, PTfACt, PTsASp, and PTsASp) result in brittle materials than can be easily broken. Poor properties with A substrates using this method are probably due to partial dissolution of cellulose acetate fibers in ethanol [29], which results in a loss of mechanical properties of the composite materials. From images from Figure 1, there is not a clear difference of the effect of using Fe(Tos)$_3$ of Fe(Trif)$_3$ as oxidizing agents. However, the effect of processing can be distinguished as processing by spray coating makes in-situ composite materials more flexible than processing by casting; this can be easily seen by comparing PTsWCt and PTfWCt with PTsWSp and PTfWSp.

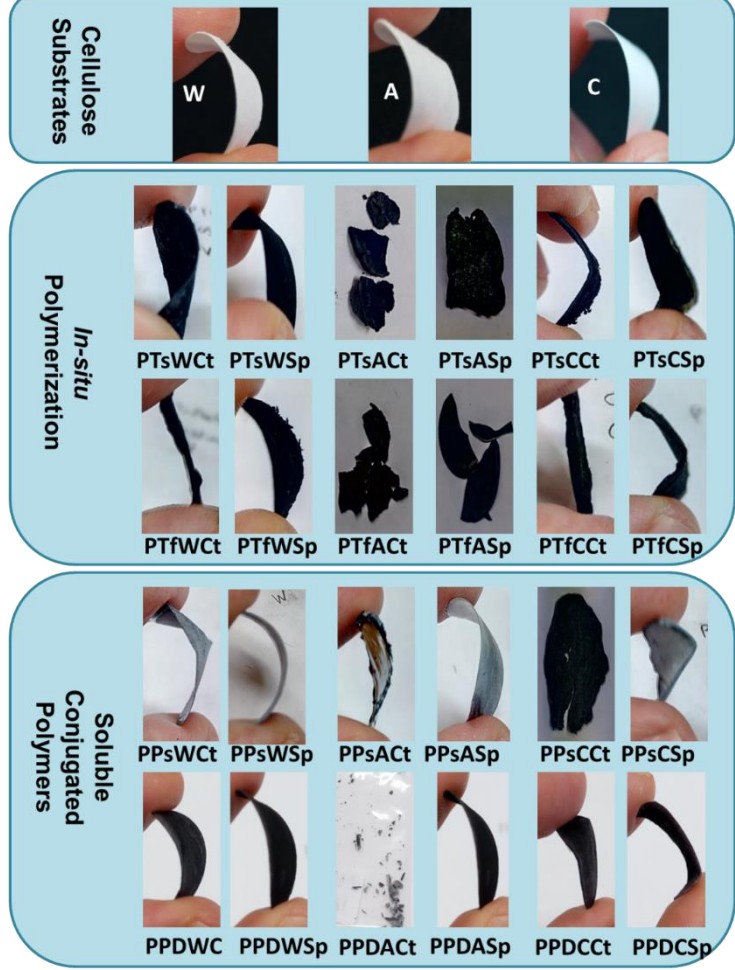

**Figure 1.** Images of polythiophene-derived composite materials.

Materials prepared from CPs solutions were prepared using a commercial PEDOT:PSS in water and PDBProDOT dissolved in DCM. When aqueous PEDOT:PSS is deposited on A membranes, they do not break or lose their properties, as in samples prepared by in-situ polymerization, indicating that cellulose acetate is not affected by water (see PPsACt and PPsASp in Figure 1). However, membranes prepared with PEDOT:PSS and W or C became brittle, probably due to swelling of fibers; this effect is more notorious when PEDOT:PSS is deposited by casting because there is a longer contact and higher interaction of the solvent with the fibers in comparison with spray coating. With exception of PPDACt, flexible composite materials were easily prepared using a dispersion of PDBProDOT in DCM. Overall, composite materials processed by spray coating have shown greater mechanical stability than materials prepared by casting. This behavior is due to the different effect of organic or aqueous solvents on the different substrates, chemically modified (A) or untreated (W/C) cellulose fibers, as well as to the effect of the oxidant used to polymerize in-situ CPs on the cellulose fibers. The presence of high concentrations of $Fe(Tos)_3$ for casting polymerization has a negative effect on the stability of the cellulose and therefore on the prepared composite materials. As shown in TGA thermal curves (Table S1, Figures S3 and S4), in all cases, the stabilities of composite materials prepared by spray coating have shown higher thermal stabilities than those prepared by casting, negatively influencing the flexibility of composite materials causing fragmentation and disintegration of final materials [30].

### 3.4. Characterization of Materials

Infrared spectroscopy. Figure 2 shows an example of FTIR-ATR spectra of PEDOT:Triflate composite material (see also Figure S5). There are no substantial differences in the spectra obtained of cellulose fibers and Whatman filter paper substrates, identifying the most intense band in the region between 1180 and 940 cm$^{-1}$ attributed to C-O and C-C stretching vibrations [31,32]. PEDOT-cellulose materials present FTIR peaks at 1518, 1350, and 1220 cm$^{-1}$ that can be attributed to the quinoid structure and stretching modes of C=C and C-C in the thiophene ring and the stretching of the C-O-C bond in the ethylendioxy group of PEDOT, which demonstrates that PEDOT was successfully coated on the cellulose fibers in all composites materials prepared [33,34]. Other characteristic peaks can be found at around 980, 830, and 695 cm$^{-1}$ due to the vibration modes of the C-S bond in the thiophene ring [35]. The FTIR spectrum of PDBProDOT (Figure S6) composite exhibits, in addition to the corresponding peaks associate to the quinoid structure and cellulose fibers as shown for PEDOT composites, the vibration peaks associated to stretching in the alkylendioxy group of the lateral chains at 1017–1048 cm$^{-1}$ [36].

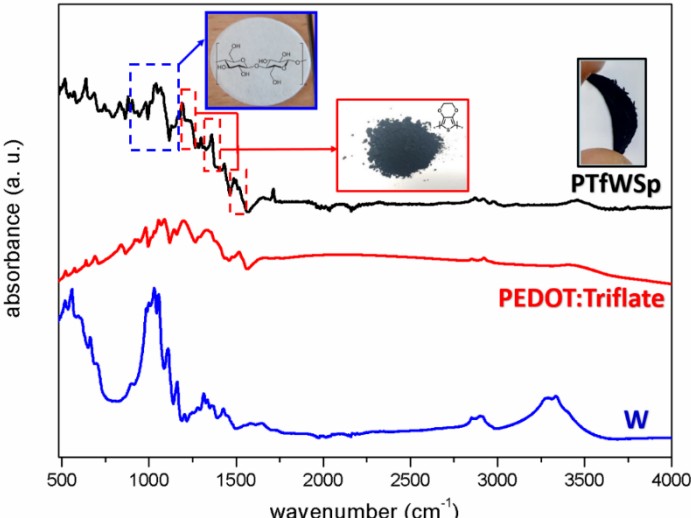

**Figure 2.** FTIR-ATR spectra of Whatman substrate (blue), PEDOT:Triflate (red), and PTfWSp composite material (black).

Cross-sectional SEM. The cross-section SEM images of in-situ composite materials were recorded in order to demonstrate the successful polymerization of PEDOT on the surface of the cellulose fibers inside the substrates. Figure 3 shows the cross-sectional micrographs of some examples of composite materials prepared in this work using different cellulosic substrates.

Electrical properties. The final purpose of these composite materials is to use them as flexible electrodes for electronic devices. Moreover, having scalability of these materials in mind, the objective of studying different approaches for the preparation of CPs-cellulose composite materials is to obtain a flexible material with good electrical properties with the lowest polymer content. Table 3 contains the results obtained of σ measured by dielectric spectroscopy (bulk conductivity) (Figures S7 and S8) and $\sigma_s$ measured by the four probe method (surface conductivity). As a reference, values for CPs are also included. Some samples previously described were not measured due to poor mechanical properties that were visually shown in Figure 1. In the case of the $\sigma_s$, some samples could not be measured due to the high resistance of the sample with the current limits proposed (1–10 mA). Table 3 also includes the polymer content in wt.%, which was calculated using the weight difference of the substrates before and after the CPs incorporation.

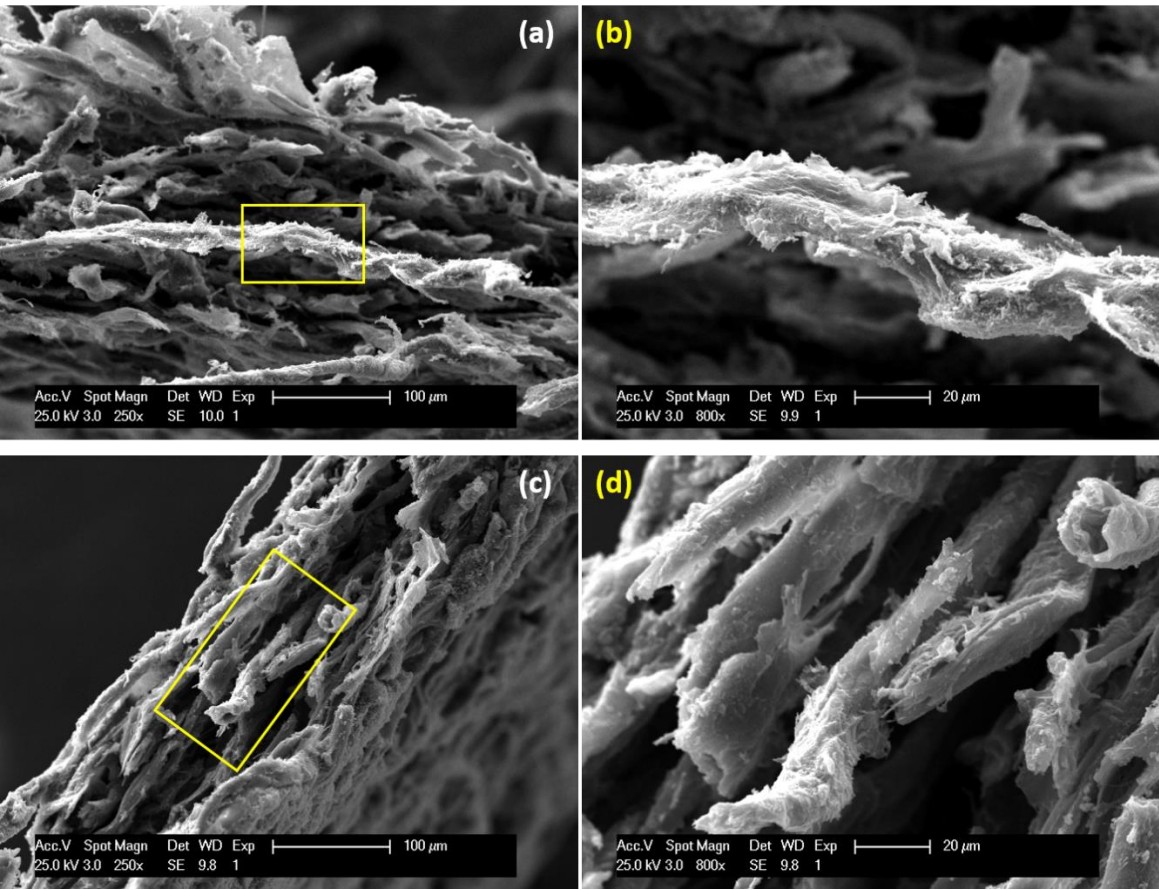

**Figure 3.** Cross-sectional SEM micrographs of (**a**,**b**) PTfCSp; and (**c**,**d**) PTfWSp. Micrographs (**a**) and (**c**) correspond at 250× magnification and micrographs (**b**,**d**) to the enlargement of the yellow boxed area (800× magnification).

The significant differences between σ and $\sigma_s$ values (Figure S9) can be explained due to the well-known anisotropy of PEDOT [37] and the influence of the heterogeneous morphology of the substrates. According to the literature, measurement of $\sigma_s$ using the four-probe method is recommended for values above $10^{-2}$ S·cm$^{-1}$; therefore, only samples with values beyond this limit were considered (Figure 4) [38]. Moreover, assessing the development of supercapacitors applications, surface conductivity suggests a more reliable and representative conductive mechanism. Thinking

about a real device, the relationship between roughness and electrical surface properties may play a fundamental role as an electrode-electrolyte interface is formed. For in-situ polymerization materials, when comparing the oxidizing agent employed, it can be seen that values obtained are similar, being slightly higher the values obtained using Fe(Trif)$_3$ in comparison with Fe(Tos)$_3$. Regarding materials prepared from CPs solutions, $\sigma_s$ value of PPsACt is higher in comparison with materials prepared by in-situ polymerization. However, this value is similar to the value obtained for pure polymer, which suggest that PEDOT is simply deposited on the surface of the substrate without penetrating the inside of the substrate. The small pore size compared with the others two substrates (0.45 μm) hinders the polymerization into the substrate, thus having both faces of the cellulosic membranes connected. Finally, though synthetized PDBProDOT demonstrated to be an easy processable polymer, using organic solvents, $\sigma_s$ of pure CPs are much lower than that of PEDOT samples.

**Table 3.** Electrical conductivity of CPs and composite materials.

| Pure Polymer Samples | $\sigma$ (S·cm$^{-1}$) | $\sigma_s$ (S·cm$^{-1}$) | Polymer content (wt.%) |
|---|---|---|---|
| *PEDOT:Tosylate* | $2.5 \times 10^{-3}$ | 8.4 | - |
| *PEDOT:Triflate* | $4.9 \times 10^{-1}$ | 6.6 | - |
| *PEDOT:PSS* | $4.5 \times 10^{-4}$ | 49 | - |
| *PDBProDOT* | $1.0 \times 10^{-3}$ | * | - |
| ***In-Situ* Polymerization** | | | |
| *PTsWCt* | $4.2 \times 10^{-5}$ | 0.05 | 43 |
| *PTsCCt* | $2.3 \times 10^{-4}$ | 0.07 | 34 |
| *PTsWSp* | $6.3 \times 10^{-6}$ | * | 29 |
| *PTsCSp* | $2.8 \times 10^{-7}$ | 0.31 | 45 |
| *PTfWCt* | $3.6 \times 10^{-4}$ | 0.60 | 80 |
| *PTfCCt* | $2.1 \times 10^{-4}$ | 0.06 | 76 |
| *PTfWSp* | $6.7 \times 10^{-5}$ | 0.46 | 24 |
| *PTfCSp* | $2.0 \times 10^{-4}$ | 0.77 | 38 |
| **Soluble CPs** | | | |
| *PPsWCt* | $6.0 \times 10^{-7}$ | * | 80 |
| *PPsACt* | $3.0 \times 10^{-5}$ | 43.67 | 68 |
| *PPsWSp* | $4.3 \times 10^{-8}$ | * | 7 |
| *PPsASp* | $1.3 \times 10^{-8}$ | * | 11 |
| *PPsCSp* | $2.5 \times 10^{-4}$ | * | 24 |
| *PPDWCt* | $5.1 \times 10^{-11}$ | * | 6 |
| *PPDCCt* | $4.2 \times 10^{-10}$ | * | 6 |
| *PPDWSp* | $1.3 \times 10^{-9}$ | * | * |
| *PPDASp* | $5.6 \times 10^{-13}$ | * | 12 |
| *PPDCSp* | $4.9 \times 10^{-11}$ | * | 4 |

* Could not be measured with current limits proposed (1–10 mA).

Surface properties. Optical 3D profilometry in combination with SEM were performed to study the surface properties of composite materials (Figure S10). Figure 5 shows a comparison between some selected materials prepared by in-situ polymerization in order to identify the effect of the processing technique and the oxidizing agent employed during preparation. Among others roughness parameters, Sa was selected and measured to give a representative value of roughness and compare for each composite material. For instance, the oxidizing agent effect is evidenced when comparing PTsWCt and PTfWCt, materials prepared under the same processing conditions but different oxidizing agent. Values $\sigma_s$ and Sa showed that the use of Fe(Trif)$_3$ allows to obtain materials with lower roughness, more homogeneous, and with a higher $\sigma_s$ than using Fe(Tos)$_3$. This dependence of roughness with surface conductivity was previously reported for thin layers of pure PEDOT [39]. A considerable effect on processing conditions is seen when comparing PTfWCt and PTfWSp, where it is observed that processing by spray coating results in a more homogeneous layer around fibers, in contrast with

casting, which creates a CP layer onto the substrate as it is not possible from micrographs to distinguish between the cellulose fibers and PEDOT. This observation was confirmed by SEM micrographs shown in the same figure. This effective covering has also a direct effect on the flexibility of materials prepared (Figure 1), which indicates that flexibility of composite materials depends on an effective coating of CPs onto fibers. Moreover, from $\sigma_s$ values, it seems that this coating around fibers allows to have similar conductivity that having a polymer layer, preserving flexibility of these materials.

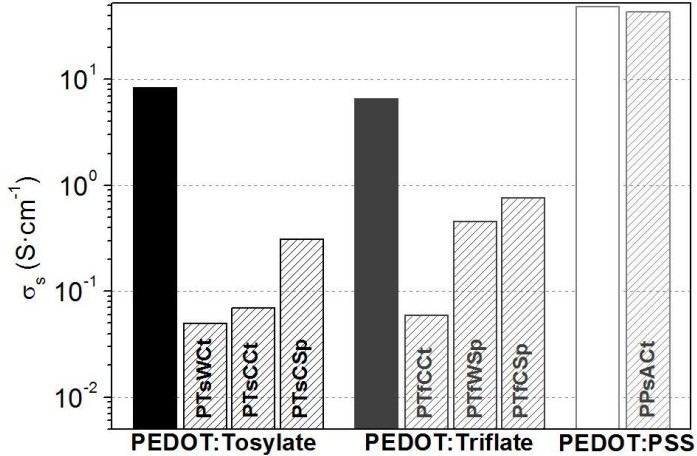

**Figure 4.** $\sigma_s$ values of raw CPs (solid bars) and selected composite materials (patterned bars).

Figure 6 shows the 3D profilometer images for selected materials prepared with CPs solutions. The effect of processing can be observed when comparing PEDOT:PSS samples deposited on A substrate. From Sa values and 3D images, it can be seen that surface of PPsACt sample is practically flat comparing to PPsASp and the substrate, confirming that a layer of PEDOT:PSS is deposited on the substrate, making this conductive coating probably not convenient for a supercapacitor electrode, where high specific area is required. PEDOT:PSS conductive layer is not formed when processing by spray coating, which results in a low $\sigma$ for PPsASp.

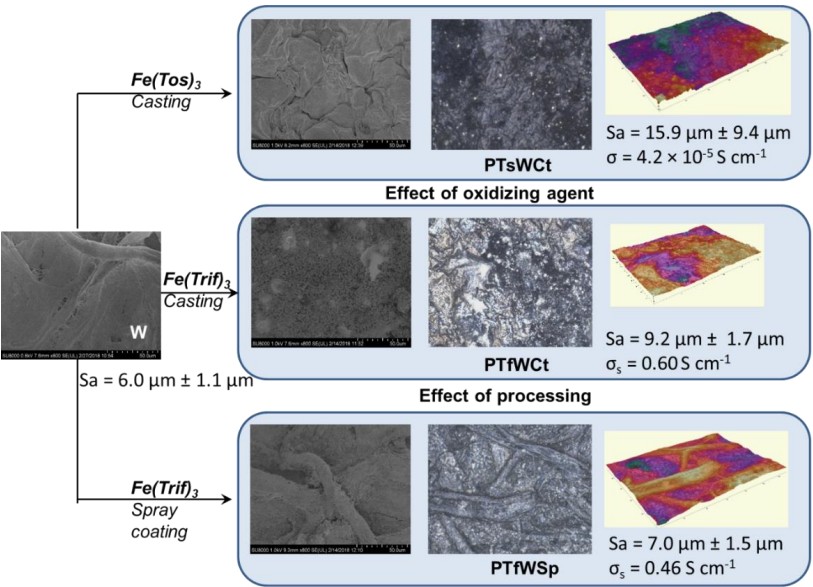

**Figure 5.** SEM micrographs and 3D profilometer images of W composites prepared by in-situ polymerization.

Regarding to PDBProDOT samples, it is observed that processing the polymer by spray coating leads to a more homogeneous coating compared to PEDOT:PSS samples; however, in the real color

image of PPDASp, some small white holes can be seen, indicating that PDBProDOT does not cover the entire substrate. When PDBProDOT is processed by spray coating over C membranes, the real color image suggests a homogeneous coating just like in-situ polymerization samples; however, despite obtaining easy-processable CPs, low σ of raw PDBProDOT leads to a low σ of composite materials. Moreover, no differences in σ can be identified independently of the substrate and roughness (PPDASp and PPDCSp). Thus, materials whose $\sigma_s$ has not been possible to measure in general show low bulk σ, which can be due to a not homogeneous coating or to a low σ of the deposited polymer. Those materials were not further studied, and only samples with a measured value of $\sigma_s$ were compared.

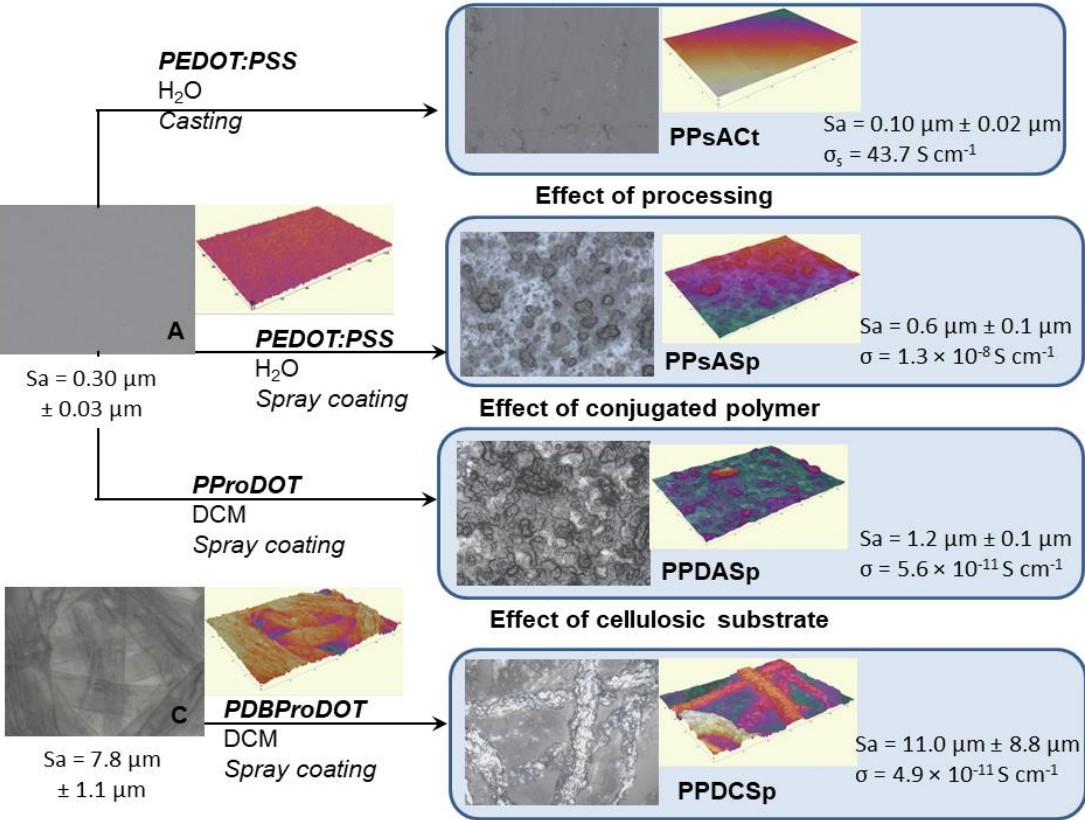

**Figure 6.** 3D Surface profilometer images of selected composite materials prepared by CPs solutions.

Figure 7 shows the dependence of $\sigma_s$ with both the PEDOT mass loading (mg·gr$^{-2}$) (Figure 7a) and Sa roughness parameter (Figure 7b and Table 4). In-situ approaches were not as satisfactory as expected for all the substrates, for instance, A substrates have a strong interaction both with the solvent used for EDOT polymerization, as well as with the oxidizing agents obtaining composite materials very brittle with very poor mechanical properties. In cases using casting technique, the obtained composite material appeared completely disintegrated (Figure 1). However, the use of CPs solutions of PEDOT:PSS or PDBProDOT with A substrates have resulted composite materials with good flexibility. Aqueous PEDOT:PSS solutions have no chemical interaction with A substrates maintaining the dimensional morphology resulting in composite materials with very low roughness independently the processing methodology (Figure S10). Moreover, a slightly decrease of $\sigma_s$ can be observed as Sa increase due to the importance of the formation of high quality crystalline phase on conductivity, which has been previously studied for PEDOT:PSS deposited onto cellulose substrates (Table 3) [40]. Likewise, PDBProDOT dissolved in low boiling point solvents (DCM) and processed by spray coating has resulted in composite materials with similar Sa values compared with PEDOT:PSS derived composite materials. In this case, casting processing working with DCM or THF has resulted in the total disintegration of the substrate discarding this methodology for the preparation of this family of composite materials.

**Table 4.** Surface conductivities, polymer content, mass loading and roughness values for in-situ prepared composite materials.

| Samples | $\sigma_s$ (S·cm$^{-1}$) | PEDOT Content (wt.%) | PEDOT Mass Loading (mg·cm$^{-2}$) | Sa (μm) |
|---|---|---|---|---|
| PTsWCt | 0.05 | 43 | 7.4 | 15.9 ± 9.5 |
| PTsCCt | 0.07 | 34 | 5.6 | 13.6 ± 3.9 |
| PTsCSp | 0.31 | 45 | 8.8 | 11.6 ± 3.1 |
| PTfWCt | 0.60 | 80 | 36.4 | 9.2 ± 1.7 |
| PTfWSp | 0.46 | 24 | 2.7 | 7.0 ± 1.5 |
| PTfCSp | 0.77 | 38 | 6.7 | 6.2 ± 0.5 |

In the case of W substrates, a main dependence of $\sigma_s$ is seen respect to the conjugated polymer used, where $\sigma_s$ of PEDOT:Triflate is generally higher than PEDOT:Tosylate composite materials, independently of Sa values. In addition to the influence of the roughness on the conductive properties, W substrates composite materials family showed a strong dependence with the intrinsic conductivity of the deposited polymers, highlighting the good performance and properties of materials prepared in-situ with PEDOT:Triflate. PTfWSp and PTfWCt showed very similar performances, Sa (7–9 μm) and improved $\sigma_s$ values (0.5–0.8 S·cm$^{-1}$) comparing with PEDOT:Tosylate derived composite materials regardless the in-situ process.

The substrates used to prepare the last family of composite materials (C) were performed *ad-hoc* in our laboratories using cellulose microfibers as detailed above. After preparing the composite materials by means of the different in-situ approaches, easily reproducible materials were obtained presenting similar $\sigma_s$ values regardless the CPs used and taking into account the significant difference in roughness covering a wide range of Sa values obtained from the profilometry (Figure S9 and Table S2). C derived composite materials Sa values showed in general higher variations than commercial substrates, which can be useful, depending of the applications, where surface parameters of these composite materials can be tuned depending of the applications (printed circuits, diodes, capacitors, supercapacitors, etc.) [14].

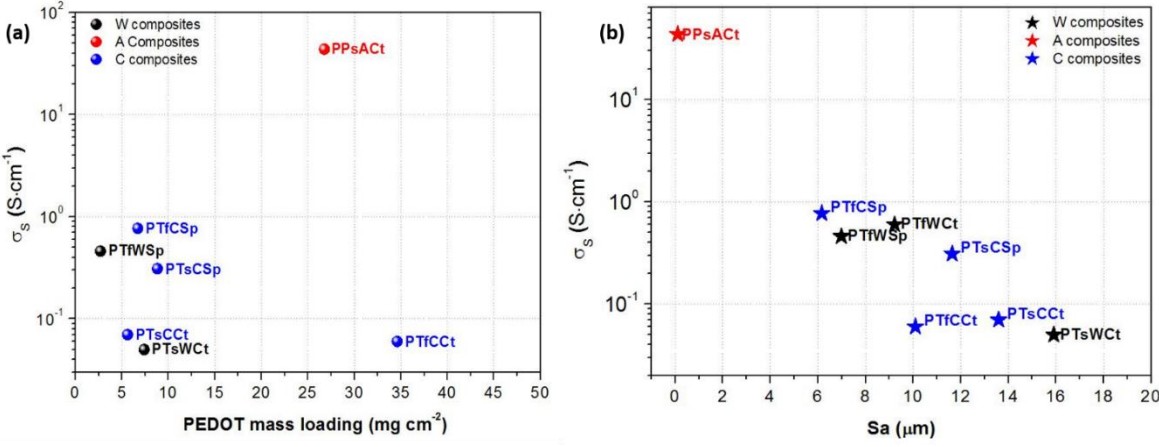

**Figure 7.** (**a**) $\sigma_s$ dependence on PEDOT mass loading; and (**b**) $\sigma_s$ dependence on Sa roughness for composite materials prepared on different substrates: Whatman® paper (black); acetate cellulose (red) and cellulose microfibers (blue).

Composite materials derived from PEDOT:PSS are induced to a surface morphological modification by the solvent, water in case, which swell cellulose fibers preventing the polymer to homogenously coat the cellulose fibers resulting in heterogeneous materials with lower σ values than expected.

As observed previously for composite materials prepared with W substrates, the use of water-based solutions does not help for a suitable processing of untreated raw cellulose substrates. In the case of PDBProDOT, the use of solutions aiding the processability of CPs allowing a slow coating formation

with longer crystallization times, as is well known the linear relationship between CPs crystallinity (percentage and quality) and conductive properties [41], is not a favorable strategy in the development of composite materials derived of PDBProDOT with improved conductivities and processability. Then, despite interaction of cellulose fibers with solvents is fundamental, processing by casting achieved a more homogeneous coatings in case of soluble polymers as PEDOT:PSS and PDBProDOT, contrary to the case of non-soluble studied polymers PEDOT:Tosylate and PEDOT:Triflate where processing by spray-coating resulted in composite materials with better performance. Then, roughness properties are more dependent mostly of the chemical and structural nature of the substrate and interactions with solvents in contrast with $\sigma$s, which are mainly influenced by the CPs used.

Hence, from Figure 7a, we can observe the non-existence of a direct correlation between PEDOT mass loading presented in each composite material with $\sigma_s$ values. As explained throughout this work, the absence of dependence is due to the influence of the material preparation on the coating of the fibers for the different cellulosic substrates. From the values shown in Table 4, no dependence is observed either between Sa parameters and PEDOT mass loadings. Composite materials derived from PEDOT:Tosylate prepared by casting possess the lower $\sigma_s$ values ($\approx 10^{-2}$ S·cm$^{-1}$) and low polymer content resulting in high values of surface roughness (Sa $\geq$ 11 μm). Similarly, PTfCCt presents $\sigma_s$ values in the same order but containing a very high PEDOT mass loading. Independently of the PEDOT source, the use of casting as processing methodology for the in-situ approach has been proven not to be the best option, due principally to the resulted heterogeneous coatings, and despite containing high polymer amounts its $\sigma_s$ is not positively affected. On the other hand, composite materials derivative of PEDOT:Triflate prepared by spray coating present $\sigma_s$ values of at least one order of magnitude higher compared with casting processing. Furthermore, the higher $\sigma_s$ values correspond to composite materials prepared with PEDOT:Triflate which corroborates the successful use of Fe(Trif)$_3$ as oxidizing agent by using this in-situ approach. Besides, this methodology has allowed to obtain flexible cellulosic composite materials with high $\sigma_s$ and low polymer content, due to effective and homogeneous coating of PEDOT:Triflate throughout the fibers of cellulose during the in-situ polymerization. Correlation displayed by Figure 7b confirms this behavior observed for PEDOT:Triflate-derived composite materials prepared by spray coating with Sa in all the selected composite materials, included for solutions of PEDOT:PSS processed by casting onto substrates of cellulose acetate (PPsACt). Moreover, an inverse trend can be appreciated for all these families of composite materials with Sa values, making a direct relationship of roughness, polymer content, and the corresponding processing conditions with electrical properties. An optimal coating of cellulose fibers during the in-situ polymerization or impregnation process is a key factor for the efficient synergy of properties; flexibility and conductivity, for all prepared composite materials. Depending on the substrate and the in-situ approach, we obtained materials with improved coating efficiency showing a continuous increase in $\sigma_s$ values (from 0.05 to 0.77 S·cm$^{-1}$) as Sa decrease (from 16 to 6 μm), despite the low PEDOT mass loading (PTfWSp contains 2.7 mg·cm$^{-2}$), and maintaining the flexibility of the cellulosic substrate unchanged. These results show the excellent relationship between PEDOT-mass loading and conductivity with the surface properties and flexibility of the composite material. The in-situ preparation of these composite materials by spray coating has achieved a synergy of properties that makes us face in a very optimistic manner the use of these materials as flexible electrodes accomplishing all the electroactive material to a pseudo-solid electrolyte succeeding a good electrochemical behavior without loss of flexibility.

## 4. Conclusions

A set polythiophene-derived composite material was prepared either by depositing CPs solutions or by performing in-situ polymerization onto three different sources of cellulose membranes using different processing methods. Two different processing approaches were evaluated to prepare in-situ composites materials: Casting and spray-coating methodologies. Spray coating presents multiple advantages respect casting, for instance, the processing time is quite short reducing the interaction solvent-substrate, which has been demonstrated that affects dramatically on mechanical properties,

flexibility, and surface morphologies of the composite material. Another advantage is the low amount of polymer needed to coat efficiently the cellulose fibers obtaining continuous, homogeneous, and reproducible coatings. Another important factor is the use of methanol as preparation and processing solvent discarding the use of more toxic and harmful to the environment solvents. Moreover, 3D optical profilometry was demonstrated as an excellent tool to characterize the surface of these successfully prepared composite materials regardless of the substrate's roughness, on the one hand, to observe the quality of the CP coatings and quantify the heterogeneity of the surface roughness (Sa factor), on the other, to correlate Sa with the electrical conductivity of composite materials. Modulating the roughness of composite materials (from 0.2 to 20 μm) by changing the source of cellulose, polymer, and/or processing conditions plays a crucial role in the resulting conductive properties of these materials. This relationship shows that the composite material prepared with PEDOT:Triflate through in-situ polymerization via spray coating results in new and versatile composite materials with promising characteristics, due to a balance between easy and fast processing, flexibility, homogeneous coatings (Sa = 6–9 μm), high conductivity $\sigma_s$ (≈0.5–0.8 S·cm$^{-1}$), and low polymer content (lower than 25 wt.%). These properties make it a scalable alternative for the preparation of flexible electrodes for energy storage applications.

**Supplementary Materials:** The following are available online at http://www.mdpi.com/2076-3417/9/16/3371/s1, Figure S1: $^1$H-NMR spectra of PDBProDOT, Figure S2: GPC traces of PDBProDOT, Figure S3: TGA curves of PEDOT:Triflate composite materials onto W substrates, Figure S4: TGA curves of polythiophene-derived composite materials, Figure S5: FTIR-ATR spectra of polythiophene-derivatives composite materials, Figure S6: FTIR-ATR spectra of PDBProDOT/cellulose microfibers composite material, Figure S7: Bulk conductivity dependence with frequency for selected cellulose composite materials obtained by in-situ polymerization, Figure S8: Bulk conductivity dependence with frequency for selected cellulose composite materials obtained by CPs solutions, Figure S9: σs and σ trends on Sa roughness, Figure S10: Sa values of raw substrates and selected composite materials, Table S1: T50, T5 and % residue for cellulose substrates and composite materials, Table S2: σ, σs and Sa of composite materials.

**Author Contributions:** F.G. and M.H. performed the data curation and formal analysis, M.H. conceived the investigation, F.G. and M.H. designed the methodology, M.H. and P.T. realized the supervision, F.G. and M.H. write the original draft, M.H. writing-review & editing, M.H. funding acquisition.

**Funding:** This research was supported by CDTI research project IDI- 20180087. We also thanks the Consejo Nacional de Ciencia y Tecnología for CVU 559770/Registro 297710 and S2013/MAE-2800.

**Conflicts of Interest:** The authors declare no conflict of interest.

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
