# Peer review of "In-Situ Approaches for the Preparation of Polythiophene-Derivative Cellulose Composites with High Flexibility and Conductivity"

_applsci, doi:10.3390/app9163371_

Round 1

Reviewer 1 Report

Dear Authors,

The manuscript titled: In-situ Approaches For the Preparation of Polythiophene-Dericative Cellulose Composites with High Flexibility and Conductivity is dealing with the preparation of a conducting polymer as a composite. The manuscript is very well presented. However, please see the comments below which should be addressed before accepting for publication:

the FTIR curves in Figure SI 5, 6 need to be replaced with more useful results. reported data on FTIR is poor in quality. A crosssectional SEM image of cellulose composite would be interesting to be added in the main text.    

my recommendation is accept after minor revisions.

Regards,

Reviewer 2 Report

This very interesting original research paper is devoted to the development of new polythiophene-derivative cellulose composites. The scientific quality of the manuscript and its importance for the field of material chemistry are undoubted. I really appreciate the amount of work performed during this research. I have found only few points which could be improved during revision of the manuscript:

In Abstract: “1 S/cm” should better be expressed as “1 S.cm-1” like in other similar cases in the manuscript. In Figure 2, Figure SI5 and Figure SI6: What is the physical quantity/unit (or some measure/scale) presented at the Y axis? There should be presented something. In Figure 5: From the statistical point of view, there should be “0.30” in “0.3 +/– 0.03” and “0.10” in “0.1 +/– 0.02”. In Figure SI2: “Log Molecular Weight” should better be expressed like at the Y axis, i.e., like “Log M”. In Figure SI5a: “-1” in “cm-1” should be in superscript. In (b): “W” should be in “wavenumber”. In Table SI2: From the statistical point of view, there should be “0.100” in “0.1 +/– 0.002”. List of references should be checked, since there are some minor imperfections in the reference style at some places.
